# *De novo* genome assembly of *Bacillus altitudinis* 19RS3 and *Bacillus altitudinis* T5S-T4, two plant growth-promoting bacteria isolated from *Ilex paraguariensis* St. Hil. (yerba mate)

Iliana Julieta Cortese[1]*, María Lorena Castrillo[1], Andrea Liliana Onetto[1], Gustavo Ángel Bich[1], Pedro Darío Zapata[1], Margarita Ester Laczeski[1,2]*

**1** Laboratorio de Biotecnología Molecular, Instituto de Biotecnología Misiones "Dra. María Ebe Reca" (InBioMis), CONICET, Facultad de Ciencias Exactas, Químicas y Naturales/FCEQyN, Universidad Nacional de Misiones/UNaM, Posadas, Misiones, Argentina, **2** Cátedra de Bacteriología, Dpto. de Microbiología, Facultad de Ciencias Exactas, Químicas y Naturales/FCEQyN, Universidad Nacional de Misiones/UNaM, Posadas, Misiones, Argentina

* cortesejulieta@gmail.com (IJC); mlaczeski@gmail.com (MEL)

## Abstract

Plant growth-promoting bacteria (PGPB) are a heterogenous group of bacteria that can exert beneficial effects on plant growth directly or indirectly by different mechanisms. PGPB-based inoculant formulation has been used to replace chemical fertilizers and pesticides. In our previous studies, two endophytic endospore-forming bacteria identified as *Bacillus altitudinis* were isolated from roots of *Ilex paraguariensis* St. Hil. seedlings and selected for their plant growth-promoting (PGP) properties shown *in vitro* and *in vivo*. The purposes of this work were to assemble the genomes of *B. altitudinis* 19RS3 and T5S-T4, using different assemblers available for Windows and Linux and to select the best assembly for each strain. Both genomes were also automatically annotated to detect PGP genes and compare sequences with other genomes reported. Library construction and draft genome sequencing were performed by Macrogen services. Raw reads were filtered using the Trimmomatic tool. Genomes were assembled using SPAdes, ABySS, Velvet, and SOAPdenovo2 assemblers for Linux, and Geneious and CLC Genomics Workbench assemblers for Windows. Assembly evaluation was done by the QUAST tool. The parameters evaluated were the number of contigs ≥ 500 bp and ≥ 1000 bp, the length of the longest contig, and the N50 value. For genome annotation PROKKA, RAST, and KAAS tools were used. The best assembly for both genomes was obtained using Velvet. The *B. altitudinis* 19RS3 genome was assembled into 15 contigs with an N50 value of 1,943,801 bp. The *B. altitudinis* T5S-T4 genome was assembled into 24 contigs with an N50 of 344,151 bp. Both genomes comprise several genes related to PGP mechanisms, such as those for nitrogen fixation, iron metabolism, phosphate metabolism, and auxin biosynthesis. The results obtained offer the basis for a better understanding of *B. altitudinis* 19RS3 and T5S-T4 and make them promissory for bioinoculant development.

**Data Availability Statement:** All relevant data are within the paper and its Supporting Information files.

**Funding:** This study was supported by Proyecto del Instituto Nacional de la Yerba Mate (INYM, Argentina) "Biofertilizantes: validación a campo y estudios de trazabilidad de la utilización de Bacillus sp. como fertilizante para yerba mate" Res. n° 274/17 (INYM - PRASY) granted to ML.

**Competing interests:** The authors have declared that no competing interests exist.

## Introduction

Biological products that enhance plant growth are an alternative for improving crop management and degraded soils. The application of native microorganisms reduces the degradation of the agroecosystem and the loss of nutrients, optimizing the yield of crops. Plant growth-promoting bacteria (PGPB) are a heterogeneous group in the rhizosphere, at root surfaces or in association with plant tissues as endophytes. These bacteria exert beneficial effects on plant growth directly, or indirectly. The mechanisms by which PGPB can influence plant growth may differ from one species to another as well as from strain to strain [1]. PGPB-based inoculant formulation and application have been used in integrated management systems to reduce or replace the use of chemical fertilizers and pesticides [2].

Bacteria genera such as *Alcaligenes*, *Acinetobacter*, *Arthrobacter*, *Azoarcus*, *Azospirillum*, *Azotobacter*, *Bacillus*, *Paenibacillus*, *Burkholderia*, *Clostridium*, *Enterobacter*, *Gluconacetobacter*, *Klebsiella*, *Kosakonia*, *Pseudomonas*, *Serratia*, and *Stenotrophomonas* include specific strains that have been reported as PGPB for different plant species [3–9]. Their mechanisms are multiple, diverse, and their effects include the transformation of nutrients into forms available to plants, for example, the capacity to fix atmospheric nitrogen [10], the solubilization of inorganic phosphorus, or the mineralization of organic phosphorus [11, 12]. Some PGPB can also promote plant growth indirectly by controlling the associated pathogens by producing antibiotics and other secondary metabolites [4], or by activating the mechanisms of Induced Systemic Resistance (ISR) [13].

Keeping in mind this context, the genus *Bacillus* presents a great diversity of species that are distributed widely in the environment. It is one of the most studied and promising genera for achieving sustainable and environmentally safe agricultural practices [14].

They have been shown to enhance plant growth through a combination of mechanisms [15, 16], by activating ISR in the plant against both root and foliar pathogens, by increasing abiotic stress tolerance [17], and by showing biocontrol properties [18]. Elicitation of ISR by *Bacillus* and its metabolites has been demonstrated on a variety of crops to defend against pathogen attack in both the greenhouse and the field [19, 20]. It can also produce numerous antifungal compounds [21], such as lipopeptides [22], bacillomycin [23], fengycin [24], surfactin [25], bacillibactin [26], and bacteriocin [27].

The PGPB activity of some bacilli strains was studied in the last years. *B. subtilis* is commercially used as a biofertilizer [28]. It can maintain stable contact with higher plants and promote their growth. *B. licheniformis* shows beneficial effects when inoculated on tomato and pepper [4]. *B. megaterium* improves different root parameters in mint [29], while *B. mucilaginosus*, when inoculated in nutrient-limited soil, can increase mineral availability [30, 31]. *B. pumilus* is used as a bioinoculant to increase the crop yield of a wheat variety in Mongolia [32]. Genome analysis revealed that *B. velezensis* can be considered a potential biofertilizer and biopesticide [33]. Likewise, *Bacillus* spp. consortia present the capability to increase the yield, growth, and nutrition of raspberry [34] and banana plants [35].

Different crops have great economic agro-food importance in the world, and it is of interest to improve their production through the implementation of PGPB as a biofertilizer [36, 37]. *Ilex paraguariensis* St. Hil., a plant that is also commonly called yerba mate, is one of the most economically important crops in the northeast of Argentina. It is widely marketed in South America, but it is also consumed worldwide. It is emphasized that despite this overall consumption, yerba mate can only grow in certain regions of Argentina, Paraguay, and Brazil due to unique soil characteristics, such as lateritic soils, and warm and moist weather [38, 39].

Currently, there are plantations of good performance in the region, however, there are concerns about the increase of degraded crop sites, as a result of the monoculture system, erosion

and compaction of soil, and nutrient loss combined with little or no soil fertilization [40]. This motivates the research and development of a biofertilizer from native bacteria isolated from yerba mate to recover crop performance.

In our previous study, two Gram-positive endophytic endospore-forming bacteria, coded as 19RS3 and T5S-T4, were isolated from roots of *I. paraguariensis* St. Hil. seedlings. These bacteria were selected for their *in vitro* PGP properties. Both strains were identified morphologically and molecularly as *Bacillus altitudinis*. Also, *B. altitudinis* 19RS3 and *B. altitudinis* T5S-T4 showed *in vivo* growth promotion in yerba mate seedlings in greenhouse conditions with promising results [41].

The study of the genome of microorganisms used for biofertilizer production is important to bioinoculant technology because it helps to identify genes that contribute to the beneficial activity and increasing knowledge of the molecular mechanisms related to plant growth potential. In the last decade, the development of new bioinformatics tools and next-generation sequencing technologies has allowed researchers to gain deeper insights into the molecular and genetic mechanisms of plant growth-promoting (PGP) activities such as the study of Pho regulon involved in the inorganic phosphate (Pi) solubilization, the detection of *nif* gene cluster associated to nitrogen fixation, the study of metabolic pathways related to the siderophore production, and the discovery of antibiotics and volatile compounds production mechanisms implicated in biocontrol properties. These advances were accompanied by an exponential increase in the number of assembler algorithms available to obtain complete prokaryotic genomes [42]. Principio del formulario Final del formulario Currently, there are two widely used classes of algorithms: overlap–layout–consensus (OLC) and de-Bruijn-graph (DBG) [43]. The DBG algorithm is based on the k-mers approach [44]. This value divides the short sequences into smaller fragments of size k, and these k-mers overlap with k-1, which represents the next k-mer. Since Illumina sequencing technology entered the global market, several short-read assembly software based on DBG have been developed, such as Velvet [45], ABySS [46], SPAdes [47], and SOAPdenovo2 [48]. Despite this, the selection of assembly tools, the determination of the parameters to be executed, as well as the evaluation of the assemblies, are still a challenge [49].

In this context, to advance knowledge of PGP mechanisms, the genomes of *B. altitudinis* 19RS3 and *B. altitudinis* T5S-T4 were sequenced. The purposes of this work were to assemble both genomes, to compare the results obtained using different *de novo* assemblers available for Windows and Linux operating systems, and to select the best assembly for each *B. altitudinis* strain. Finally, both genomes were automatically annotated to detect genes involved in PGP capabilities and compare these sequences with other *Bacillus* sp. genomes reported.

## Materials and methods

### Bacteria

*B. altitudinis* 19RS3 and *B. altitudinis* T5S-T4 were isolated from roots of *I. paraguariensis* St. Hil. seedlings [41]. Both strains were identified by analysis of 16S rRNA gene sequencing (accession number MH883312 and MH883235, respectively) and characterized as Gram-positive endospore-forming rod-shaped bacteria. *B. altitudinis* 19RS3 and *B. altitudinis* T5S-T4 were deposited into the bacterial collection of the Instituto de Biotecnología Misiones "Dra. María Ebe Reca", under accession numbers LBM250 and LBM251, respectively. Bacteria were preserved in 50% glycerol stocks at -80˚C until the performance of this study.

### DNA extraction

The strains were cultivated in nutrient broth (Britania Lab. SA) for 24 h at 30˚C. The DNA extraction procedures were done using Sambrook´s work protocol modified [50, 51]. The

DNA was resuspended by 20 μL of sterile distilled DNAse-free water (BioPack ®). The extracted DNA was qualitatively evaluated by agarose gel (1% w/v) electrophoresis stained with a solution of GelRed® (Sigma-Aldrich, Germany). The DNA quantification was performed by UV spectrophotometry.

## Library preparation and genome sequencing

Genomic TruSeq Nano DNA library (350) construction and draft genome paired-end sequencing were performed by Macrogen Co. (Seoul, Korea) services using Illumina HiSeq technology.

## Genome assembly and evaluation

The quality of the FASTQ files was verified with FastQC [52] and reads were trimmed to ensure high quality (Phred score > 30) using Trimmomatic version 0.39 [53].

The genomes were assembled using different *de novo* assemblers available for Linux and Windows operating systems (Table 1).

The k-mer values were selected according to the manual user instructions of each assembler. In general terms, the values were odd to avoid palindromes and were strictly inferior to read length.

The assemblies obtained in Linux were evaluated using QUAST (Quality Assessment Tool for Genome Assemblies) [56–58]. The assemblies generated in Windows showed their own statistics tables.

The parameters evaluated were the number of contigs ≥ 500 bp, the number of contigs ≥ 1000 bp, the length of the longest contig, and the value of N50.

## Genome annotation

Gene prediction and annotation were performed using The Rapid Prokaryotic Genome Annotation (Prokka) [59]. Putative genes involved in plant growth-promoting mechanisms were determined using the Rapid Annotations using Subsystems Technology (RAST) [60] annotation server and KEGG Automatic Annotation Server (KAAS) [61].

## Genome comparison

The genomes of *B. altitudinis* 19RS3, *B. altitudinis* T5S-T4, *B. altitudinis* W3 (accession number: NZ_CP011150.1) an NCBI reference sequence, *B. altitudinis* GQYP101 (accession

**Table 1. *De novo* assemblers used for the genome assemblies of *Bacillus altitudinis* 19RS3 and *Bacillus altitudinis* T5S-T4 plant growth-promoting bacteria isolated from *Ilex paraguariensis* St. Hil.**

| Assembler | Reference | Operating System | Manual |
|---|---|---|---|
| Velvet (v. 1.2.10) | Zerbino et al. [45] | Linux | https://github.com/dzerbino/velvet/wiki/Manual |
| ABySS (v. 2.0.2) | Simpson et al. [46] | Linux | ftp://ftp.ccb.jhu.edu/pub/dpuiu/Docs/ABYSS.html |
| SPAdes (v. 3.12.0) | Bankevich et al. [47] | Linux | http://cab.spbu.ru/files/release3.12.0/manual.html |
| SOAPdenovo2 (v. 2.40) | Luo et al. [48] | Linux | https://vcru.wisc.edu/simonlab/bioinformatics/programs/soap/SOAPdenovo2MANUAL.txt |
| Geneious (v. 11.0.1) | Kearse et al. [54] | Windows | https://assets.geneious.com/documentation/geneious/GeneiousManual.pdf |
| CLC Genomics Workbench (v. 12.0.3) | Knudsen et al. [55] | Windows | http://resources.qiagenbioinformatics.com/manuals/clcgenomicsworkbench/900/index.php?manual=Sequence_alignment.html |

number: NZ_CP040514.1) an NCBI reference sequence reported as PGPB, and *B. velezensis* FZB42 (accession number: CP000560.2) [23] a commercial PGPB strain used as an active principle of Biomex® and Rhizovital®42, were compared to locate genes involved in PGP mechanisms using Geneious 11.0.1 software.

## Results

The genome sequencing of *B. altitudinis* 19RS3 showed 9,938,250 paired-end reads of 101 bp with a GC content of 41.014% and an average coverage of 249. While *B. altitudinis* T5S-T4 showed 12,397,272 paired-end reads of 101 bp with a GC content of 40.390% and an average coverage of 292.

After the quality filtering by Trimmomatic, *B. altitudinis* 19RS3 and *B. altitudinis* T5S-T4 resulted in 9,329,838 and 10,923,782 paired-end reads, respectively.

Genome assembly's quality statistics generated by different assemblers for *B. altitudinis* 19RS3 and *B. altitudinis* T5S-T4 are summarized in Tables 2 and 3, respectively. The complete quality statistics obtained by all the assemblers using different k-mer values are available as S1–S12 Tables.

In the assembly of *B. altitudinis* 19RS3 genome, ABySS and Velvet generated the contigs with the highest N50 value. These two assemblies produced N50 values that are more than five times higher than the worst assemblies. Velvet also generated the fewest number of contigs and performed considerably better than the other assemblers. Geneious generated the worst assembly with the fewest N50 value and the highest number of contigs.

For the assembly of *B. altitudinis* T5S-T4 genome, ABySS had the highest N50 value, followed by Velvet. This last assembler also generated the fewest number of contigs. The CLC Genomics Workbench assembly, despite its large N50 contig size, had more contigs than any other assembler.

The best assembly for both genomes was obtained using the Velvet software. The *B. altitudinis* 19RS3 genome was assembled into 15 contigs ($\geq$ 500 bp) with an N50 value of 1,943,801 bp and the longest contig length of 1,943,801 bp. The *B. altitudinis* T5S-T4 genome was assembled into 24 contigs ($\geq$ 500 bp) with an N50 of 344,151 bp and the longest contig length of 805,135 bp. The *B. altitudinis* 19RS3 and *B. altitudinis* T5S-T4 assembled contigs were deposited in Genbank under accession numbers JACAAH01 and JACAAI01 respectively.

Genomic features of *B. altitudinis* 19RS3 (Fig 1) and *B. altitudinis* T5S-T4 (Fig 2) presented similar size, noncoding sequences, ribosomal RNA sequences, and transfer RNA (Table 4).

**Table 2. Comparison of assembled genome quality statistics generated by different assemblers for *Bacillus altitudinis* 19RS3 a plant growth-promoting bacterium isolated from *Ilex paraguariensis* St. Hil.**

| Assembler | SPAdes | ABySS | Velvet | SOAPdenovo2 | Geneious (Velvet) | CLC Genomics Workbench |
|---|---|---|---|---|---|---|
| K-mer | 79 | 87 | 93 | 89 | 91 | 64 |
| # contigs ($\geq$ 500 bp) | 16 | 18 | 15 | 43 | - | 43 |
| # contigs ($\geq$ 1000 bp) | 14 | 14 | 12 | 39 | 37 | 18 |
| Largest contig (bp) | 966.271 | 1.184.276 | 1.943.801 | 492.824 | - | 966.324 |
| N50 (bp) | 931.914 | 928.348 | 1.943.801 | 227.675 | 155.382 | 895.161 |

K-mer: k value used to execute the assemblers.

# contigs $\geq$ 500 bp: number of contigs larger or equal to 500 bp.

# contigs $\geq$ 1000 bp: number of contigs larger or equal to 1000 bp.

N50: length of the contig overlapping the midpoint of the length-order concatenation of contigs.

**Table 3. Comparison of assembled genome quality statistics generated by different assemblers for *Bacillus altitudinis* T5S-T4 a plant growth-promoting bacterium isolated from *Ilex paraguariensis* St. Hil.**

| Assembler | SPAdes | ABySS | Velvet | SOAPdenovo2 | Geneious (Velvet) | CLC Genomics Workbench |
|---|---|---|---|---|---|---|
| K-mer | 97 | 95 | 89 | 91 | 89 | 20 |
| # contigs (≥ 500 bp) | 34 | 28 | 24 | 61 | - | 79 |
| # contigs (≥ 1000 bp) | 31 | 22 | 23 | 51 | 44 | 74 |
| Largest contig (bp) | 805.153 | 807.963 | 805.135 | 656.867 | - | 534.586 |
| N50 (bp) | 344.108 | 552.057 | 344.151 | 131.452 | 222.742 | 214.550 |

K-mer: k value used to execute the assemblers.

# contigs ≥ 500 bp: number of contigs larger or equal to 500 bp.

# contigs ≥ 1000 bp: number of contigs larger or equal to 1000 bp.

N50: length of the contig overlapping the midpoint of the length-order concatenation of contigs.

In the chromosome sequence of *B. altitudinis* 19RS3 a total of 3861 CDSs and 80 RNAs genes were predicted (Table 4). Among these CDSs 2762 (68.43%) genes were classified into 469 functional subsystems. Similarly, in the chromosome sequence of *B. altitudinis* T5S-T4 a total of 3801 CDSs and 67 RNAs genes were predicted (Table 4). Among these CDSs 2750 (69.54%) genes were classified into 472 functional subsystems. There is a high similarity among the number of genes in each category between *B. altitudinis* 19RS3 and *B. altitudinis* T5S-T4; but the former has more genes in several metabolism-related functions in cellular processes such as cell wall formation and capsule formation, regulation and cellular signaling, and genes related to phages, prophages, transposable elements, plasmids (Table 5). Most of the genes were associated with the metabolism of carbohydrates and amino acids derivates.

The presence of related genes to PGPR mechanisms or the metabolic pathway prediction of RAST was found from the gene annotation. The production of enzymes involved in the metabolism of indole acetic acid (IAA) via the tryptophan pathway coded by the gene cluster *trp*(ABD) was predicted, suggesting that *B. altitudinis* 19RS3 and *B. altitudinis* T5S-T4 have the potential to biosynthesize auxin. The gene cluster that encodes to produce bacilibactin, *dhb*(ACEBF), was also found in both genomes showing the potential for the production of siderophore.

The *pst*(SCAB) genes, coding for Pi-specific transporter, were found in the genome of *B. altitudinis* 19RS3 and B. *altitudinis* T5S-T4 suggesting the capacity of both strains for inorganic phosphate solubilization. Finally, the genes *nif*(U) and *nif*(S), were present in both strains which are involved in nitrogenase enzymatic activity responsible for the biological fixation of nitrogen. However, the presence of the complete gene cluster which is essential for the nitrogenase activity was not found.

Volatile compounds as 2,3-butanediol and acetoin might be produced by *B. altitudinis* 19RS3 and *B. altitudinis* T5S-T4 given that it has the potential to produce the enzymes α−acetolactate synthetase, α−acetolactate decarboxylase, and acetoin utilization protein. Coding regions for surfactin production were also found and the complete gene cluster *srf*(ABCD) was annotated in each genome. Genes responsible for flagellar motility, chemotaxis, and biofilm synthesis, which allow *B. altitudinis* 19RS3 and T5S-T4 to move toward root-exudates facilitating adhesion to plant surfaces, were encountered. Also, some genes related to stress response, such as implicated in osmotic stress, oxidative stress, cold and heat shock, and detoxification, in addition to genes related to sporulation were present in both genomes, indicating a possible protection mechanism to extreme environmental conditions.

The genomes comparison revealed specific gene clusters involved in PGP capabilities (Table 6). All the genomes presented genes associated with the production of volatile

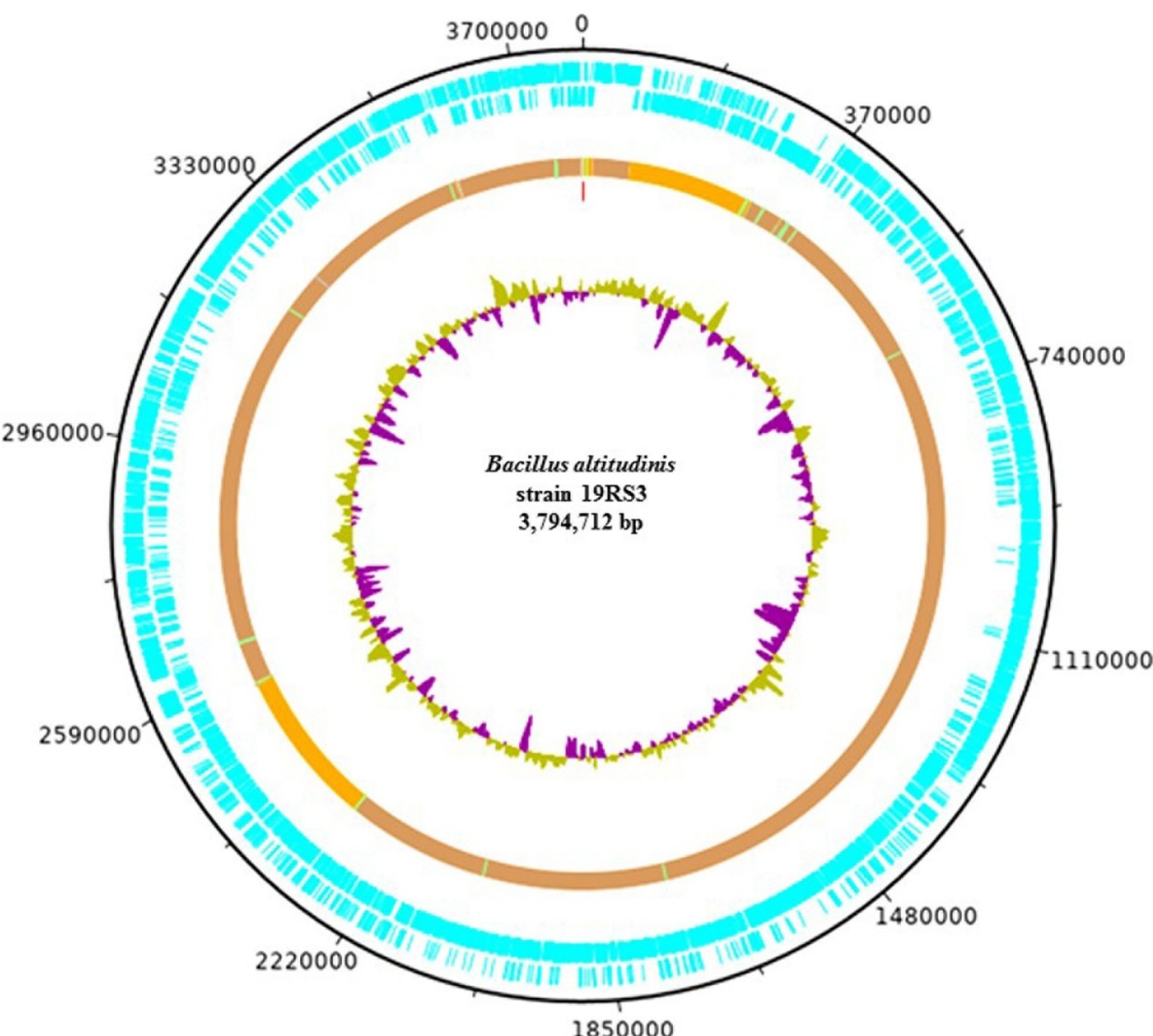

**Fig 1. The chromosomal organization of *Bacillus altitudinis* 19RS3 a plant growth-promoting bacterium isolated from *Ilex paraguariensis* St. Hil.**
Circularized DNA plotter diagram of the chromosome of *B. altitudinis*, orientated from the origin; the outer black circle designates the genome base positions around the chromosome. The outer blue circles depict predicted 3861 CDSs on both the forward and reverse strands. The predominantly brown circle represents the main chromosomal core structure with likely horizontally acquired DNA elements, including areas representing non-coding RNA (ncRNA) and areas representing tRNA. The inner circle is a GC skew plot [GC/(G+C)].

compounds such as 2,3-butanediol and acetoin. Only *B. altitudinis* 19RS3, *B. altitudinis* T5S-T4 and *B. velezensis* FZB42 showed the presence of genes associated with surfactin production. This commercial strain also presented genes for phytase and iturin production. Interestingly, other genes coding for bacilibactin, IAA production, Pi-specific transporter, and PHO regulon were discovered only in our studied strains.

## Discussion

In the present study the genome of two PGP strains isolated from *I. paraguariensis* St. Hil., *B. altitudinis* 19RS3 and *B. altitudinis* T5S-T4, were sequenced and assembled. We compared the

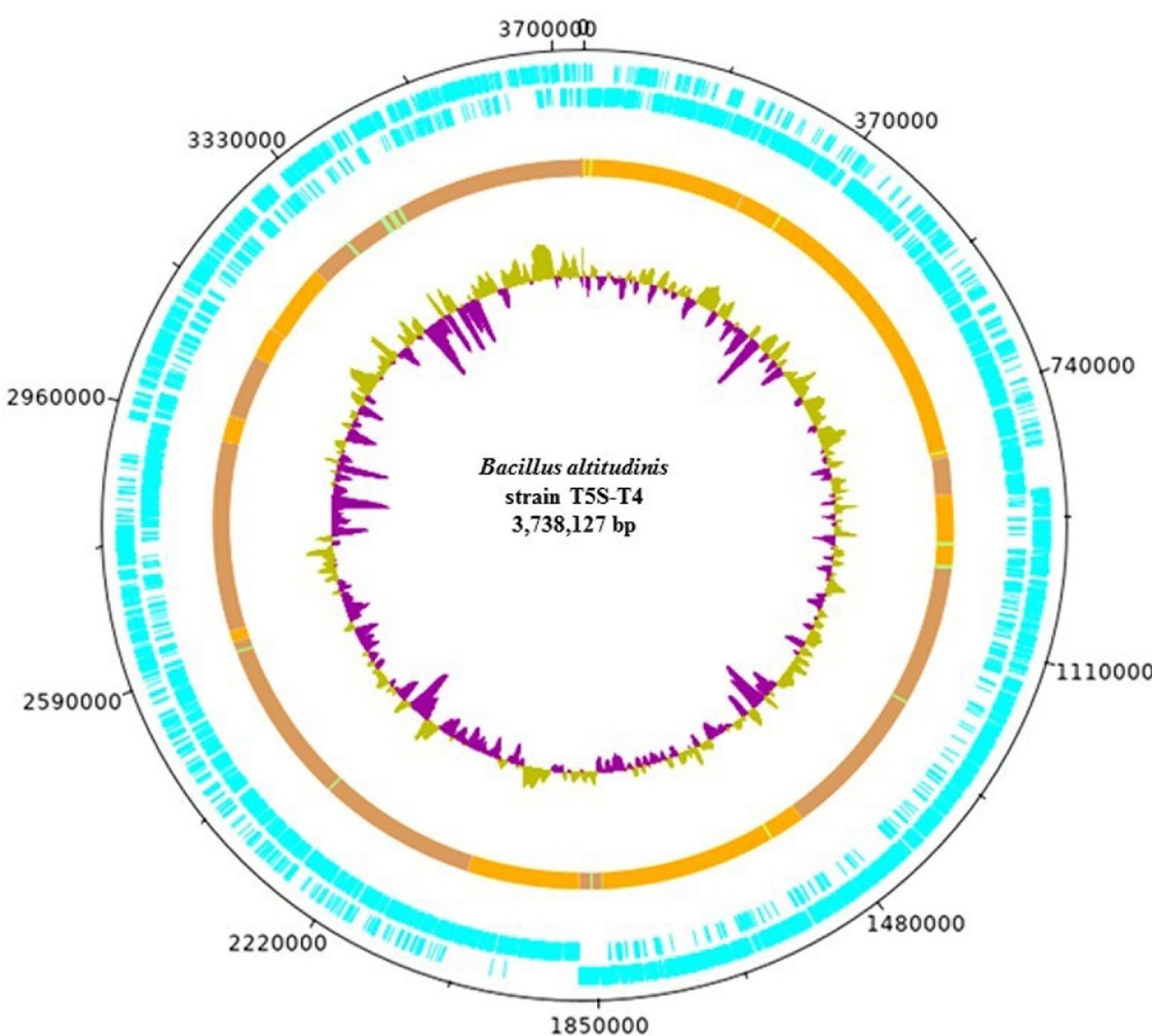

**Fig 2. The chromosomal organization of *Bacillus altitudinis* T5S-T4 a plant growth-promoting bacterium isolated from *Ilex paraguariensis* St. Hil.** Circularized DNA plotter diagram of the chromosome of *B. altitudinis*, orientated from the origin; the outer black circle designates the genome base positions around the chromosome. The outer blue circles depict predicted 3801 CDSs on both the forward and reverse strands. The predominantly brown circle represents the main chromosomal core structure with likely horizontally acquired DNA elements, including areas representing non-coding RNA (ncRNA) and areas representing tRNA. The inner circle is a GC skew plot [GC/(G+C)].

assembled genome quality statistics generated by different *de novo* assemblers available for Windows and Linux operating systems. Although no assembler was the best in all the various metrics simultaneously, the Velvet assembler generated the fewest contig number and the higher N50 value. We also annotated both genomes, detected the genes associated with PGP properties, and determinate the presence of these sequences in two *B. altitudinis* genomes reported in the NCBI.

The prokaryotic genomic structure characteristics were considered to select the sequencing platform, as well as the construction of the library. Some authors [62] indicate it may be useful to try different strategies for *de novo* assembly of a newly sequenced organism. They propose

**Table 4. General genome features of *Bacillus altitudinis* strain 19RS3 and T5S-T4 plant growth-promoting bacteria isolated from *Ilex paraguariensis* St. Hil.**

| Features | 19RS3 chromosome | T5S-T4 chromosome |
|---|---|---|
| Genome size | 3,794,712 | 3,738,127 |
| G + C (%) | 41.2 | 41.2 |
| Predicted CDS | 3861 | 3801 |
| rRNAs | 6 | 7 |
| tRNAs | 74 | 60 |
| Genbank accession | JACAAH01 | JACAAI01 |

C+G (%): guanine and cytosine content; CDS: protein-coding genes; rRNAs: ribosomal RNA; tRNAs: transfer RNA.

to evaluate the strategies for the construction of contigs and analyze their effect on the assembly when choosing the best parameters. They also emphasize that knowing the characteristics of the genomic structure of an organism, the sequencing platform, and the construction of the library can be especially useful when choosing assembly tools.

The raw reads obtained for both genomes were processed to eliminate adapters and possible contaminants that can affect the quality of the results, creating a problem when comparing the efficiency of the assemblers. Some authors [28] recommend a trimming step to ensure the

**Table 5. Annotation of genes involved in the metabolism and other cellular processes of *Bacillus altitudinis* 19RS3 and *Bacillus altitudinis* T5S-T4 plant growth-promoting bacteria isolated from *Ilex paraguariensis* St. Hil.**

| Genes functions | | *B. altitudinis* 19RS3 | *B. altitudinis* T5S-T4 |
|---|---|---|---|
| Genes related to metabolism | Fatty acids, lipids, and isoprenoids | 114 | 112 |
| | Amino acids derivatives | 434 | 426 |
| | Sulfur | 34 | 34 |
| | Carbohydrates | 447 | 444 |
| | Cofactors, vitamins, prosthetic groups, pigments | 195 | 196 |
| | Aromatics compounds | 6 | 8 |
| | DNA | 74 | 100 |
| | Phosphorus | 26 | 26 |
| | Iron | 48 | 47 |
| | Secondary metabolism | 4 | 4 |
| | Nitrogen-proteins | 277 | 262 |
| | Nucleosides and nucleotides | 109 | 109 |
| | Potassium | 7 | 7 |
| | RNA | 153 | 154 |
| Genes related to cellular processes | Division and cellular cycle | 58 | 58 |
| | Dormancy and sporulation | 120 | 120 |
| | Cellular wall and capsule formation | 152 | 148 |
| | Photosynthesis | 0 | 0 |
| | Miscellaneous | 40 | 40 |
| | Motility and chemostasis | 87 | 88 |
| | Regulation and cellular signaling | 67 | 59 |
| | Related to phages, prophages, transposable elements, plasmids | 20 | 14 |
| | Respiration | 64 | 64 |
| | Response to stress | 93 | 95 |
| | Membrane transport | 78 | 78 |
| | Virulence, disease, and defense | 55 | 57 |

**Table 6. Comparative genomics of *Bacillus altitudinis* 19RS3 and *Bacillus altitudinis* T5S-T4 with the reported genomes of *Bacillus altitudinis* W3, *Bacillus altitudinis* GQYP101 and *Bacillus velezensis* FZB42.**

| Compound | Genes | Gen (kpb) | *B. altitudinis* 19RS3 | *B. altitudinis* T5S-T4 | *B. altitudinis* W3 | *B. altitudinis* GQYP101 | *B. velezensis* FZB42 |
|---|---|---|---|---|---|---|---|
| Bacilibactin | *dhb* (ACEBF) | 11.7 | + | + | - | - | - |
| Surfactin | *srf*(ABCD) | 26.2 | + | + | - | - | + |
| 2,3 butanediol dehydrogenase | | 1.04 | + | + | + | + | + |
| Acetoin utilization protein | *acu*(C) | 1.16 | + | + | + | + | + |
| Acetolactate decarboxylase | *bud*(A) | 0.77 | + | + | + | + | + |
| Acetolactate synthase | *als*(S) | 1.7 | + | + | + | + | + |
| IAA production | *trp*(ABD) | 3.02 | + | + | - | - | - |
| Pi-specific transporter | *pst*(SCAB) | 3.54 | + | + | - | - | - |
| PHO regulon | *pho*(RP) | 2.47 | + | + | - | - | - |
| Phytase | *phy*(C) | 1.15 | - | - | - | - | + |
| Iturin | *bmy*(DABC) | 37.2 | - | - | - | - | + |

+ presence of the complete gene cluster

- absence of the complete gene cluster

high quality of the data. We agree and highlight the importance of filtering and trimming to generate better results because in a previous study we evaluated the effect of the use of raw and filtered reads as input files, in the assembly of the genome of *B. altitudinis* 19RS3 and obtained a better assembly using the filtered reads [63].

When considering the number of contigs, the longest contig length, and the N50 value in the assemblies of both genomes, the software Velvet and ABySS generated the best results. As in our study, other authors [64] evaluated *de novo* assemblers using reads of prokaryotic genomes. In their work, Velvet showed a greater number of contigs and a lower value of N50, while ABySS generated a lower number of contigs in the paired data sets and showed a higher N50 value. The authors associated these results variation to factors such as the quality of the data and the k-mer size. About this last item, we decided to use different k-mer values considering their effect in genome assemblies. For SPAdes assembler, we used a k-mer value of 63 greater than the average size of the reads and we sought to gradually increase the values, getting to obtain more precise assemblies with k-mer values of 79 and 97. Large k-mers often result in larger contigs, but excessively large k-mers can cause a fragmented graph with a higher number of contigs. SPAdes, ABySS, and SOAPdenovo2 generated their best assemblies with the highest k-mer value, however, they also produced the most fragmented assemblies. Several authors [65, 66], showed SPAdes stands out as one of the best assemblers for the assembly of Illumina data, due to its quality and high precision. Although in our study, assemblies with a fewer number of contigs were obtained using other software, SPAdes produced very good results for the assembly of both genomes.

As showed in the assemblies obtained in this work, the value that presented the greatest variation was the number of contigs. We agree with some authors [49] that the wide variety of assemblers' available use different heuristic approaches to meet the challenges of genome assembly and this results in significant differences when comparing the number of contigs they generated. For this reason, we consider necessary a thorough and complete evaluation of the assembled genome quality statistics generated by different assemblers before selecting the best assembly.

In the present study, we predicted genes and enzymes associated with PGP mechanisms in the *B. altitudinis* 19RS3 and *B. altitudinis* T5S-T4 genomes. We detected genes related to the conversion pathway of tryptophan to indole, which is consistent with the determined indole production observed in the *in vitro* assays [41]. The presence of the bacillibatin gene cluster showed the potential of siderophore production, while the detection of Pi transporters and the Pho regulon indicated a possibility for inorganic phosphate solubilization. The presence of *nif* (U) and *nif*(S) was also determined in both genomes suggesting the possibility of the strain to fix environmental nitrogen. The properties mentioned above are consistent with the *in vitro* and *in vivo* PGP activities determined experimentally in previous studies for both strains [41].

The results obtained for the assembly of *B. altitudinis* 19RS3 and *B. altitudinis* T5S-T4 genomes are like the reported for other *Bacillus* PGP strains such as *B. flexus* KLBMP 4941 [15–67], *B. pumilus* GM3FR [68], *B. mycoides* GM6LP [69], *B. vallismortis* NBIF-001 [70], *B. velezensis* 2A-2B [71], and *B. velezensis* UCMB5140 [14]. Particularly the genome of *B. altitudinis* FD48 [72] comprises several genes related to plant growth promotion mechanisms, such as those for the biogenesis of organic acids involved in inorganic phosphorus solubilization, iron, and siderophore uptake systems, and nitrogen metabolism. Perhaps of this, genome annotation isn´t available to realize a deeper genome comparison. The PGP genes reported for *B. subtilis* EA-CB0575 [28] related to IAA, siderophore production, acetoin, 2,3-butanediol, and LPs production, nitrogen fixation, and phosphate solubilization are like those detected in *B. altitudinis* 19RS3 and *B. altitudinis* T5S-T4 genomes. The comparison realized with *B. altitudinis* W3, *B. altitudinis* GQYP101, and *B. velezensis* FZB42 genomes indicated that our strains present some unique genes able to promote *I. paraguariensis* growth. The five genomes present genes associated with the production of volatile compounds as 2,3-butanediol and acetoin, but the other PGP gene clusters were only detected in our studied strains. Also, we determinate the presence of loci for surfactins codification in the genomes of *B. altitudinis* 19RS3, *B. altitudinis* T5S-T4, and *B. velezensis* FZB42. The commercial strain FZB42 also presents genes to the phytase and iturin production. Each *Bacillus* PGP strain provides a subtle difference in terms of their plant growth-promoting and biocontrol activities. Future design of an effective bioinoculant should be based on combinations of PGP strains supplementing each other.

## Conclusion

The best assembly for *B. altitudinis* 19RS3 and *B. altitudinis* T5S-T4 was obtained using the Velvet software. A great number of genes associated with PGP mechanisms were annotated and analyzed. It was found genes involved in auxin biosynthesis, siderophore production, phosphate metabolism, and nitrogen fixation. Also, other genes associated with acetoin and 2,3-butanediol production, motility, chemotaxis, adhesion, sporulation, and defense functions were encountered. The gene detection realized in the present study supports the PGP properties observed in previous assays.

The results obtained offer the basis for a better understanding of *B. altitudinis* 19RS3 and T5S-T4 biology and make them promissory for the development of novel strategies in the biotechnological application of these bacteria as bioinoculant. The information presented here will allow in-depth functional and comparative genome analyses to provide a better understanding of beneficial plant-bacteria associations.

## Supporting information

**S1 Table. Assembled genome quality statistics obtained for *Bacillus altitudinis* 19RS3 a plant growth-promoting bacterium isolated from *Ilex paraguariensis* St. Hil. using ABySS**

**assembler.**
(DOCX)

**S2 Table. Assembled genome quality statistics obtained for *Bacillus altitudinis* T5S-T4 a plant growth-promoting bacterium isolated from *Ilex paraguariensis* St. Hil. using ABySS assembler.**
(DOCX)

**S3 Table. Assembled genome quality statistics obtained for *Bacillus altitudinis* 19RS3 a plant growth-promoting bacterium isolated from *Ilex paraguariensis* St. Hil. using SPAdes assembler.**
(DOCX)

**S4 Table. Assembled genome quality statistics obtained for *Bacillus altitudinis* T5S-T4 a plant growth-promoting bacterium isolated from *Ilex paraguariensis* St. Hil. using SPAdes assembler.**
(DOCX)

**S5 Table. Assembled genome quality statistics obtained for *Bacillus altitudinis* 19RS3 a plant growth-promoting bacterium isolated from *Ilex paraguariensis* St. Hil. using Velvet assembler.**
(DOCX)

**S6 Table. Assembled genome quality statistics obtained for *Bacillus altitudinis* T5S-T4 a plant growth-promoting bacterium isolated from *Ilex paraguariensis* St. Hil. using Velvet assembler.**
(DOCX)

**S7 Table. Assembled genome quality statistics obtained for *Bacillus altitudinis* 19RS3 a plant growth-promoting bacterium isolated from *Ilex paraguariensis* St. Hil. using SOAP-denovo2 assembler.**
(DOCX)

**S8 Table. Assembled genome quality statistics obtained for *Bacillus altitudinis* T5S-T4 a plant growth-promoting bacterium isolated from *Ilex paraguariensis* St. Hil. using SOAP-denovo2 assembler.**
(DOCX)

**S9 Table. Assembled genome quality statistics obtained for *Bacillus altitudinis* 19RS3 a plant growth-promoting bacterium isolated from *Ilex paraguariensis* St. Hil. using geneious assembler with the Velvet algorithm.**
(DOCX)

**S10 Table. Assembled genome quality statistics obtained for *Bacillus altitudinis* T5S-T4 a plant growth-promoting bacterium isolated from *Ilex paraguariensis* St. Hil. using geneious assembler with the Velvet algorithm.**
(DOCX)

**S11 Table. Assembled genome quality statistics obtained for *Bacillus altitudinis* 19RS3 a plant growth-promoting bacterium isolated from *Ilex paraguariensis* St. Hil. using CLC workbench assembler.**
(DOCX)

**S12 Table. Assembled genome quality statistics obtained for *Bacillus altitudinis* T5S-T4 a plant-growth-promoting bacteria isolated from *Ilex paraguariensis* St. Hil. using CLC workbench assembler.**
(DOCX)

## Author Contributions

**Conceptualization:** María Lorena Castrillo, Margarita Ester Laczeski.

**Data curation:** Iliana Julieta Cortese, Andrea Liliana Onetto, Gustavo Ángel Bich, Margarita Ester Laczeski.

**Formal analysis:** Iliana Julieta Cortese, Gustavo Ángel Bich, Margarita Ester Laczeski.

**Funding acquisition:** Margarita Ester Laczeski.

**Investigation:** Iliana Julieta Cortese, María Lorena Castrillo, Andrea Liliana Onetto, Gustavo Ángel Bich, Margarita Ester Laczeski.

**Methodology:** María Lorena Castrillo, Gustavo Ángel Bich, Margarita Ester Laczeski.

**Project administration:** Margarita Ester Laczeski.

**Resources:** Margarita Ester Laczeski.

**Software:** María Lorena Castrillo.

**Supervision:** María Lorena Castrillo, Pedro Darío Zapata, Margarita Ester Laczeski.

**Validation:** Margarita Ester Laczeski.

**Visualization:** Gustavo Ángel Bich, Margarita Ester Laczeski.

**Writing – original draft:** Iliana Julieta Cortese, María Lorena Castrillo, Gustavo Ángel Bich, Margarita Ester Laczeski.

**Writing – review & editing:** Iliana Julieta Cortese, Margarita Ester Laczeski.

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
