## [Decision Letter · Decision Letter 0]

16 Dec 2020

PONE-D-20-28962

De novo assembly of *Bacillus altitudinis* 19RS3 and *Bacillus
altitudinis* T5S-T4, two Plant Growth-Promoting Bacteria isolated from
*Ilex paraguariensis St. Hil. (yerba mate)*

PLOS ONE

*Dear Dr. *Laczeski*,*

Thank you for submitting your manuscript to PLOS ONE. After careful
consideration, we feel that it has merit but does not fully meet PLOS ONE’s
publication criteria as it currently stands. Therefore, we invite you to submit
a revised version of the manuscript that addresses the points raised during the
review process.

Please submit your revised manuscript by January 2. If you will need more
time than this to complete your revisions, please reply to this message or
contact the journal office at plosone@plos.org. When you're ready to submit your revision, log on
to https://www.editorialmanager.com/pone/ and select the
'Submissions Needing Revision' folder to locate your manuscript
file.

If you would like to make changes to your financial disclosure, please
include your updated statement in your cover letter. Guidelines for resubmitting
your figure files are available below the reviewer comments at the end of this
letter.

We look forward to receiving your revised manuscript.

Kind regards,

Ying Ma, Ph.D.

Academic Editor

PLOS ONE

Journal Requirements:

"The authors would like to thank the Consejo Nacional de Investigaciones Científicas
y Técnicas (CONICET, Argentina) for Julieta Cortese and Andrea Onetto’s doctoral
fellowships. Gustavo Bich is a postgraduate CONICET fellowship holder. Lorena
Castrillo, Margarita Laczeski and Pedro Zapata are CONICET assistant, associate and
independent researchers, respectively."

 Proyecto del Instituto Nacional de la Yerba Mate (INYM, Argentina)
"Biofertilizantes: validación a campo y estudios de trazabilidad de la utilización
de Bacillus sp. como fertilizante para yerba mate” Res. nº 274/17 (INYM -
PRASY).

Reviewers' comments:

Reviewer's Responses to Questions

**Comments to the Author**

1. Is the manuscript technically sound, and do the data support the
conclusions?

Reviewer #1: Yes

Reviewer #2: Partly

2. Has the statistical analysis been performed
appropriately and rigorously? 

Reviewer #1: Yes

Reviewer #2: N/A

3. Have the authors made all data underlying the
findings in their manuscript fully available?

Reviewer #1: Yes

Reviewer #2: No

4. Is the manuscript presented in an
intelligible fashion and written in standard English?

Reviewer #1: No

Reviewer #2: Yes

5. Review Comments to the Author

Reviewer #1: This manuscript describes the de novo sequencing and assembly of
genomes from two isolates of Bacillus altitudinis that were recovered from plant
tissue. The authors have used various computational tools to obtain the best
assembly, and have annotated the genomes of both isolates. Part of the
annotation identified putative functions that may be related to the plant growth
promotion capabilities of the isolates. The sequencing and analysis is pretty
routine, but the results are pretty clear and this provides helpful information
to the body of information of genome analysis of plant growth promoting
rhizobacteria (PGPR). I have two important points to be addressed, plus a few
minor comments, as follows:

1. The paper would be strengthened by examining the relationship between
these two new genomes with other published Bacillus genomes, especially those
with PGPR activity. When compared, what genes are conserved or are novel? This
would be helpful to know in order to more closely connect potential genes with
PGPR mechanisms.

2. Related to the above, the current analysis identifies broad categories of
genes potentially involved in PGPR activity, such as iron metabolism, but some
more in depth analysis of specific genes would be helpful. In particular,
identify if genes are present that have been identified in other Bacillus
species.

3. Line 130, what tissues were used for DNA isolation? Leaves,
etc.?

4. Please have someone carefully proofread for proper English, including use
of 'a', 'an', and 'the'. Often one of these words is present when not needed, or
absent when it is needed. See for example in lines 38, 73 and 86.

5. Line 32 should read 'Assembly evaluation was done...'

6. line 66, should read '...great diversity of species...', 'Kepping' should
be 'Keeping'

7. line 68, 'perspective' seems to be the wrong word here.

8. line 82, should be 'revealed'

9. lines 105, 106, remove 's' from the ends of words where it is not
needed.

10. line 142, 'in' should be 'by'

11. line 176, 'bigger' is an awkward word here, better to say 'higher' or
'larger'

12. lines 181-183, the end of the sentence after the last comma is redundant
and should be removed.

13. line 190 and table, are the accession numbers really all zeros after the
initial letters, or are these placeholders to be updated?

Reviewer #2: General comments: This manuscript described the assembly and
annotation of two plant growth-promoting bacteria. The authors describe multiple
different assemblies constructed using various different software programs and
choose the assembly they believe is the best. This is an interesting dataset and
reporting that could be of interest to bacterial researchers and researchers
interested in using more natural means of beneficials for plant growth and
overall health.

Manuscript concerns:

1. One main concern of the manuscript is what this adds to the community.
Many previous studies have looked at bacterial genome assemblies across multiple
software types. It seems for the most part this manuscript agrees with basically
all previous findings. The authors should really focus and point out what their
results are adding to the community. Adding the assemblies of PGPB is great and
justified, but the manuscript focuses so much on the assembly of multiple
softwares that it dilutes down the importance of having these genomes without
really adding much to the space of assembly software decisions.

2. The methods are insufficient in details. It would be really difficult for
anyone to reproduce your results and assemblies. We are not told the types of
reads used, how each of the assembly software was run and utilized (what
parameters were used or changed or prioritized).

3. Something isn’t right with the T5S-T4 CLC workbench results in Table 3.
The largest contain is 178bp but the N50 is 895bp?

4. I’m questioning the availability of the data for this manuscript. The only
thing available seems to be contigs assemblies from Velvet. Where is the rest of
the data used and generated in the manuscript that would be useful to the
community? As far as I can tell the raw read information isn’t available and
either is the annotations done for the assemblies. These exclusions don’t really
adhere to the data availability guidelines of the journal.

5. Why are there such differences in the assemblies? It seems reasonable for
one to think that the T5S-T4 genome might have a better assembly as there is
more input data, but the findings show the opposite with the T5S-T4 genome
having substantially more contigs and much smaller N50. Is there an underlying
data difference, characteristic of the genome, or other possible reason for
this?

6. For the T5S-T4 assembly, why was the Velvet assembly picked as the best
over the ABySS assembly? Going by metrics of # of contigs it is very close and
the ABySS assembly has a larger contig and a much larger N50.

7. It seems a bit more could be done to choose which assembly is the ‘best’
than just contig number, largest contig size, and N50. Other measures that could
be considered would be map the reads back to the assemblies to determine the
number that align and at what mapping quality. Also, the annotation is not
assessed at all, but could also be used to help determine the assembly quality
by using a BUSCO or similar software to check for gene completeness. These would
really help strengthen the reasoning for picking an assembly over
others.

8. Figure 3 and 4: Is there are better way the authors can think of to
present this data? The pie chart does’t really add anything and it is difficult
to match up colors of categories and the chart. The authors could perhaps at
least order the output and categories by size or some other manner for easier
comparisons for the readers.

9. pg 4 line 88-89: What region are you referring too? The reader isn’t
familiar with where you are.

10. pg 5 line 108-11: What type of insights? Could expand this to help the
reader understand what is known in the genomics of PGPB to see how your work is
important and fits and adds to the community.

11. pg 14 line 249-250: There is no highlighting the importance of filtering
in this manuscript. There are no results of assemblies without trimming to
compare to assemblies with trimming, so this statement can’t really be made
based on the data presented in this manuscript.

12. pg 14 line 259-262: This sentence is confusing. It is unclear how kmers
were picked and how you would use kmers that are larger than some, or most, of
the read lengths. If the kmer length was larger then the read was the read
removed from the analysis?

13. Grammar and tense usage throughout the manuscript needs to be
checked.

6. PLOS authors have the option to publish the
peer review history of their article (what does this mean?). If published, this
will include your full peer review and any attached files.

If you choose “no”, your identity will remain anonymous but your review may
still be made public.

**Do you want your identity to be public for this peer review?** For
information about this choice, including consent withdrawal, please see our
Privacy
Policy.

Reviewer #1: No

Reviewer #2: No

---

## [Author Response · Author response to Decision Letter 0]

12 Jan 2021

Response to reviewers

Title: De novo genome assembly of Bacillus altitudinis 19RS3 and Bacillus altitudinis
T5S-T4, two Plant Growth-Promoting Bacteria isolated from Ilex paraguariensis St.
Hil. (yerba mate)

Authors: Iliana Julieta Cortese, María Lorena Castrillo, Andrea Liliana Onetto,
Gustavo Ángel Bich, Pedro Darío Zapata, Margarita Ester Laczeski.

First, we want to express our sincere thanks to the reviewers for their great work,
their notes have allowed us not only to improve the manuscript but also to reflect
on future research.

After a careful review of our proposed article, based on the suggestions made by the
reviewers, we have processed your submission for a new evaluation. In the new
manuscript, we have used the track changes mode in Word for the modifications made
to the original text as you suggested.

We hope that the work done will achieve the final approval of the Editorial Team. If
not, all authors are available to resolve any issue or proceed with further
revisions, as necessary.

We then respond to each of the reviewers' observations.

Reviewer Comments:

Reviewer 1

This manuscript describes the de novo sequencing and assembly of genomes from two
isolates of Bacillus altitudinis that were recovered from plant tissue. The authors
have used various computational tools to obtain the best assembly,and have annotated
the genomes of both isolates. Part of the annotation identified putative functions
that may be related to the plant growth promotion capabilities of the isolates. The
sequencing and analysis is pretty routine, but the results are pretty clear and this
provides helpful information to the body of information of genome analysis of plant
growth promoting rhizobacteria (PGPR). I have two important points to be addressed,
plus a few minor comments, as follows:

1. The paper would be strengthened by examining the relationship between these two
new genomes with other published Bacillus genomes, especially those with PGPR
activity. When compared, what genes are conserved or are novel? This would be
helpful to know in order to more closely connect potential genes with PGPR
mechanisms

Response: Following your suggestion we have incorporated in the manuscript the
conserved genes of both bacterial strains related to PGP properties.

2. Related to the above, the current analysis identifies broad categories of genes
potentially involved in PGPR activity, such as iron metabolism, but some more in
depth analysis of specific genes would be helpful. In particular, identify if genes
are present that have been identified in other Bacillus species. 

Response: As suggested, new relationships between these two new genomes and other
published Bacillus genomes, especially of bacteria with PGPR activity, were
examined. A table was added to add information and improve analysis in the results
section. In addition, related information was added to the discussion.

3. Line 130, what tissues were used for DNA isolation? Leaves, etc.?

Response: We changed tissues for roots.

4. Please have someone carefully proofread for proper English, including use of 'a',
'an', and 'the'. Often one of these words is present when not needed, or absent when
it is needed. See for example in lines 38, 73 and 86. 

Response: English proofread was checked

5. Line 32 should read 'Assembly evaluation was done...' 

Response: We changed the text.

6. line 66, should read '...great diversity of species...', 'Kepping' should be
'Keeping'

Response: We corrected the word.

7. line 68, 'perspective' seems to be the wrong word here.

Response: The expression has been improved.

8. line 82, should be 'revealed'

Response: We changed the text.

9. lines 105, 106, remove 's' from the ends of words where it is not needed.

Response: We removed “s” from the ends of words where it wasn´t needed.

10. line 142, 'in' should be 'by'

Response: We changed the word.

11. line 176, 'bigger' is an awkward word here, better to say 'higher' or 'larger' 

Response: We change bigger to higher.

12. lines 181-183, the end of the sentence after the last comma is redundant and
should be removed.

Response: We removed the sentence.

13. line 190 and table, are the accession numbers really all zeros after the initial
letters, or are these placeholders to be updated?

Response: We thank the reviewer for this particularly important remark. We corrected
the accession numbers of both genomes.

Reviewer 2

General comments: This manuscript described the assembly and annotation of two plant
growth-promoting bacteria. The authors describe multiple different assemblies
constructed using various different software programs and choose the assembly they
believe is the best. This is an interesting dataset and reporting that could be of
interest to bacterial researchers and researchers interested in using more natural
means of beneficials for plant growth and overall health.

1. One main concern of the manuscript is what this adds to the community. Many
previous studies have looked at bacterial genome assemblies across multiple software
types. It seems for the most part this manuscript agrees with basically all previous
findings. The authors should really focus and point out what their results are
adding to the community. Adding the assemblies of PGPB is great and justified, but
the manuscript focuses so much on the assembly of multiple softwares that it dilutes
down the importance of having these genomes without really adding much to the space
of assembly software decisions.

Response: We thank the Reviewer 2 for her/his careful reading of the manuscript and
for her/his constructive remarks, which were useful in improving our paper. We
followed this important comment. In the new version of our manuscript, we added more
information about the PGP genes detected in both genomes. We considered this
information should be useful for future comparative genome analyses to provide a
better understanding of beneficial plant-bacteria associations.

2. The methods are insufficient in details. It would be really difficult for anyone
to reproduce your results and assemblies. We are not told the types of reads used,
how each of the assembly software was run and utilized (what parameters were used or
changed or prioritized).

Response: We added the type of reads used. About the parameters applied for each
software we used the pre-determinate options indicated in the manual user
instructions and we only vary the k-mer value. In the new version of our manuscript,
we include the genome quality statistics generated by different assemblers as
Supplementary material.

3. Something isn’t right with the T5S-T4 CLC workbench results in Table 3. The
largest contain is 178bp but the N50 is 895bp?

Response: We checked and corrected the values.

4. I’m questioning the availability of the data for this manuscript. The only thing
available seems to be contigs assemblies from Velvet. Where is the rest of the data
used and generated in the manuscript that would be useful to the community? As far
as I can tell the raw read information isn’t available and either is the annotations
done for the assemblies. These exclusions don’t really adhere to the data
availability guidelines of the journal.

Response: We report in Genbank all the information related to this project:
Bioproject, Biosample, and WGS data, which are the focus of our manuscript and put
this information available to the community. In the Genbank server, we considered
reporting only the best data of assemblies (contigs from Velvet) to avoid errors or
confusion. Also, related to this query, in the results section, we included a
sentence making explicit the availability of the assembled genome. Now we also add
as supporting information the other assemblies statistics obtained in this work. 

As for the raw sequence information, we are still working with that data. We
appreciate your understanding of their reserve until we finish with their
processing. If more information is needed for the scientific community at this
stage, we ask the reviewers to convey the concern again.

5. Why are there such differences in the assemblies? It seems reasonable for one to
think that the T5S-T4 genome might have a better assembly as there is more input
data, but the findings show the opposite with the T5S-T4 genome having substantially
more contigs and much smaller N50. Is there an underlying data difference,
characteristic of the genome or other possible reason for this?

Response: Although the genome of B. altitudinis T5S-T4 has more reads, it doesn´t
mean that it will generate a better assembly. The data of both genomes were obtained
in the same way, and the characteristics of the reads generated were remarkably
similar, so we don´t identified any underlying difference.

6. For the T5S-T4 assembly, why was the Velvet assembly picked as the best over the
ABySS assembly? Going by metrics of # of contigs it is very close and the ABySS
assembly has a larger contig and a much larger N50.

Response: About this item, we decided to select the assembly with a lower number of
contigs ≥ 500 bp as better. Velvet assembler is commonly used in prokaryotic genome
assembly, so we take the decision mentioned above. 

7. It seems a bit more could be done to choose which assembly is the ‘best’ than just
contig number, largest contig size, and N50. Other measures that could be considered
would be map the reads back to the assemblies to determine the number that align and
at what mapping quality. Also, the annotation is not assessed at all, but could also
be used to help determine the assembly quality by using a BUSCO or similar software
to check for gene completeness. These would really help strengthen the reasoning for
picking an assembly over others.

Response: As we answered in the commentary 2, in the new version of our manuscript we
include the genome quality statistics generated by different assemblers as
Supplementary material.

8. Figure 3 and 4: Is there are better way the authors can think of to present this
data? The pie chart does’t really add anything and it is difficult to match up
colors of categories and the chart. The authors could perhaps at least order the
output and categories by size or some other manner for easier comparisons for the
readers.

Response: The information from Figures 3 and 4 were replaced and reorganized in Table
5 for better understanding.

9. pg 4 line 88-89: What region are you referring too? The reader isn’t familiar with
where you are

Response: We clarified this sentence by adding: Ilex paraguariensis St. Hil., a plant
that is also commonly called yerba mate, is one of the most economically important
crops in the northeast of Argentina.

10. pg 5 line 108-11: What type of insights? Could expand this to help the reader
understand what is known in the genomics of PGPB to see how your work is important
and fits and adds to the community

Response: We added more information about the molecular and genetic mechanisms of
plant growth promoting (PGP) activities.

11. pg 14 line 249-250: There is no highlighting the importance of filtering in this
manuscript. There are no results of assemblies without trimming to compare to
assemblies with trimming, so this statement can’t really be made based on the data
presented in this manuscript

Response: A citation of a previous work of our authorship was included in the
discussion section to add information related to trimming and filtering steps and
their effect on the genome assembly of B. altitudinis 19RS3. In that publication, we
compare the assemblies obtained by using raw-reads and filtered-reads as input
files.

12. pg 14 line 259-262: This sentence is confusing. It is unclear how kmers were
picked and how you would use kmers that are larger than some, or most, of the read
lengths. If the kmer length was larger then the read was the read removed from the
analysis?

Response: We added more information about the k-mer size selection in the materials
and methods section. In general terms, the values were odd to avoid palindromes and
were strictly inferior to read length.

13. Grammar and tense usage throughout the manuscript needs to be checked.

Response: We thank Reviewers 1 and 2 for their careful reading of the manuscript and
for their constructive remarks, which were useful in improving our paper. We
followed this important comment and the manuscript writing was completely
checked.

to Reviewers.docx
---

## [Decision Letter · Decision Letter 1]

24 Feb 2021

De novo assembly of *Bacillus altitudinis*19RS3 and *Bacillus
altitudinis* T5S-T4, two Plant Growth-Promoting Bacteria isolated from
*Ilex paraguariensis St. Hil. (yerba mate)*

PONE-D-20-28962R1

*Dear Dr. *Laczeski*,*

We’re pleased to inform you that your manuscript has been judged
scientifically suitable for publication and will be formally accepted for
publication once it meets all outstanding technical requirements.

Kind regards,

Ying Ma, Ph.D.

Academic Editor

PLOS ONE

Additional Editor Comments (optional):

Reviewers' comments:

Reviewer's Responses to Questions

**Comments to the Author**

1. If the authors have adequately addressed your comments raised in a
previous round of review and you feel that this manuscript is now acceptable for
publication, you may indicate that here to bypass the “Comments to the Author”
section, enter your conflict of interest statement in the “Confidential to
Editor” section, and submit your "Accept"
recommendation.

Reviewer #1: All comments have been addressed

Reviewer #2: All comments have been addressed

2. Is the manuscript technically sound, and do
the data support the conclusions?

Reviewer #1: Yes

Reviewer #2: Yes

3. Has the statistical analysis been performed
appropriately and rigorously? 

Reviewer #1: Yes

Reviewer #2: Yes

4. Have the authors made all data underlying the
findings in their manuscript fully available?

Reviewer #1: Yes

Reviewer #2: No

5. Is the manuscript presented in an
intelligible fashion and written in standard English?

Reviewer #1: No

Reviewer #2: Yes

6. Review Comments to the Author

Reviewer #1: Thank you for addressing the concerns mentioned in my previous
review. The manuscript is considerably improved and I feel it now provides for a
better comparison with other Bacillus strains with PGPB activity. There are
still places in the text where the English or specific wording could be
improved.

Line 72, genres should be genera

Line 116, delete the s to read biofertilizer production. There are other
places where there is an unneeded s at the end of a word, and other similar word
issues throughout.

Reviewer #2: Comments: I want to thank the authors for making improvements on
most of the comments from the reviews. The revisions that where made improve the
manuscript greatly. The authors did a commendable job of taking the reviews and
improving the substance, structure, and digestibility of the
manuscript.

7. PLOS authors have the option to publish the
peer review history of their article (what does this mean?). If published, this
will include your full peer review and any attached files.

If you choose “no”, your identity will remain anonymous but your review may
still be made public.

**Do you want your identity to be public for this peer review?** For
information about this choice, including consent withdrawal, please see our
Privacy
Policy.

Reviewer #1: No

Reviewer #2: No

---

## [Editor Report · Acceptance letter]

1 Mar 2021

PONE-D-20-28962R1 

*De novo* genome assembly of *Bacillus altitudinis*
19RS3 and *Bacillus altitudinis* T5S-T4, two Plant Growth-Promoting
Bacteria isolated from *Ilex paraguariensis* St. Hil. (yerba mate) 

Dear Dr. Laczeski:

I'm pleased to inform you that your manuscript has been deemed suitable for
publication in PLOS ONE. Congratulations! Your manuscript is now with our
production department. 

Kind regards, 

on behalf of

Dr. Ying Ma 

Academic Editor

PLOS ONE